

**Modulation of the seasonal cycle of the Antarctic sea ice extent by sea ice**
**processes and feedbacks with the ocean and the atmosphere**
Hugues Goosse[1], Sofia Allende Contador[1], Cecilia M. Bitz[2], Edward Blanchard-Wrigglesworth[2], Clare
Eayrs[3], Thierry Fichefet[1], Kenza Himmich[4], Pierre-Vincent Huot[5], François Klein[1], Sylvain Marchi[5],
François Massonnet[1], Bianca Mezzina[1], Charles Pelletier[6], Lettie Roach[7,8], Martin Vancoppenolle[4],
Nicole P.M. van Lipzig[5]
1. Earth and Life Institute, Université catholique de Louvain, Belgium
2. Department of Atmospheric Sciences, University of Washington, Seattle, USA
3. Center for global Sea Level Change, New York University Abu Dhabi, United Arab Emirates
4. Sorbonne Université, Laboratoire d'Océanographie et du Climat (LOCEAN-IPSL), CNRS, IRD,

12        MNHN, Paris, France

5. Department of Earth and Environmental Sciences, KU Leuven, Leuven, Belgium
6. European Centre for Medium-Range Weather Forecasts, Bonn, Germany
7. NASA Goddard Institute for Space Studies, New York, NY, USA
8. Center for Climate Systems Research, Columbia University, New York, NY, USA
Corresponding author: Hugues Goosse hugues.goosse@uclouvain.be





**Abstract**

The seasonal cycle of the Antarctic sea ice extent is strongly asymmetric, with a relatively slow increase
after the summer minimum followed by a more rapid decrease after the winter maximum. This cycle
is intimately linked to the seasonal cycle of the insolation received at the top of the atmosphere but
sea ice processes as well as the exchanges with the atmosphere and ocean may also play a role. To
quantify these contributions, a series of idealized sensitivity experiments have been performed with
an eddy-permitting (1/4°) NEMO-LIM3 Southern Ocean configuration including a representation of ice
shelf cavities, in which the model was either driven by an atmospheric reanalysis or coupled to the
COSMO-CLM$^2$ regional atmospheric model. In those experiments, sea ice thermodynamics and
dynamics as well as the exchanges with the ocean and atmosphere are strongly perturbed. This is
achieved by modifying snow and ice thermal conductivities, the vertical mixing in the ocean top layers,
the effect of freshwater uptake/release upon sea ice growth/melt, ice dynamics and surface albedo.
We find that the evolution of sea ice extent during the ice advance season is largely independent of
the direct effect of the perturbation and appears thus mainly controlled by initial state in summer and
subsequent insolation changes. In contrast, the melting rate varies strongly between the experiments
during the retreat, in particular if the surface albedo or sea ice transport are modified, demonstrating
a strong contribution of those elements to the evolution of ice coverage through spring and summer.
As with the advance phase, the retreat is also influenced by conditions at the beginning of the melt
season in September. Atmospheric feedbacks enhance the model winter ice extent response to any of
the perturbed processes, and the enhancement is strongest when the albedo is modified. The response
of sea ice volume and extent to changes in entrainment of subsurface warm waters to the ocean
surface is also greatly amplified by the coupling with the atmosphere.

**Short Summary**  (500 characters)

Using idealized sensitivity experiments with a regional atmosphere-ocean-sea ice model, we show that
the sea ice advance is constrained by initial conditions in March while the retreat season is influenced
by the magnitude of several physical processes, in particular by the ice-albedo feedback and ice
transport. Atmospheric feedbacks amplify the response of the winter ice extent to perturbations while
some negative feedbacks related to heat conduction fluxes act on the ice volume.



## 1. Introduction

The sea ice extent in the Southern Ocean, defined as the ocean surface covered by at least 15% of sea ice, displays a very pronounced seasonal cycle with a minimum in February of about 3 million $km^2$ and a maximum in September of more than 18 million $km^2$ on average over the last decades (Parkinson, 2014, 2019; Handcock and Raphael, 2020) (Fig. 1). In contrast to the Arctic, where multiyear ice accounted for a significant fraction of the total ice extent -at least until the end of the 20[th] century-, the Antarctic sea ice cover is mainly seasonal, with sea ice only present in summer in some regions close to the coast, in particular in the Weddell and Ross Seas.

The seasonal cycle of Antarctic sea ice extent is highly asymmetric, with a minimum around Julian day 50 (February 19) and a maximum on average close to day 260 (September 18) (Stammerjohn et al., 2008; Massom et al., 2013; Handcock and Raphael, 2020; Raphael et al., 2020; Roach et al., 2022). The advance season, defined as the time between the minimum and maximum ice extents, is thus about two months longer than the retreat season, defined as the time from maximum to minimum.

It has been suggested that this asymmetry is related to the variations of the mean position of the westerly winds that blow over the Southern Ocean associated with the Semi Annual Oscillation (SAO) (Enomoto and Ohmura, 1990; Watkins and Simmonds, 1999; Eayrs et al., 2019). This mode of variability of the Antarctic climate induces a larger divergence of the sea ice pack in spring and thus a rapid melting, while the divergence is weaker in autumn, leading to a slower expansion of the pack. A complementary mechanism explaining the rapid seasonal retreat of the sea ice is the positive ice-albedo feedback, in which a decrease in ice concentration yields a larger absorption of solar radiation and enhances the ice melting (Gordon, 1981; Nihashi and Cavalieri, 2006). A possible role of the oceanic heat input has also been proposed (Gordon, 1981). However, the vertical ocean heat transport from the relatively warm ocean below the mixed layer to the surface is higher in autumn and winter when the stratification is weak than in spring and summer when it is strong (Gordon, 1981; Martinson, 1990). The seasonality of the vertical oceanic transport alone could thus not explain the asymmetry in the seasonal cycle of the sea ice extent (Eayrs et al., 2019), but it could have an indirect effect, for instance through its effect on the ice thickness (Martinson, 1990; Goosse et al., 2018; Wilson et al., 2019).

Nevertheless, a recent study based on idealized climate models has demonstrated that the asymmetry of the seasonal cycle of the ice extent is due to the seasonal cycle of incoming solar radiation (Roach et al., 2022). The period with relatively high incoming solar radiation in spring and summer induces a rapid melting season and a fast retreat of the sea ice, while a long period with low insolation in autumn and winter favors a longer growing season. This relatively direct mechanism is very robust and explains why the asymmetry is observed each year and is reproduced by a wide range of models, from very simple ones to the most complex Earth System models (Eayrs et al., 2019; Roach et al., 2022).

Identifying the seasonal cycle of insolation as the main contributor to the asymmetry of the seasonal cycle of the Antarctic sea ice extent is a major achievement. However, the atmosphere, sea ice and ocean dynamics still play a role and may modulate the magnitude of the asymmetry. Furthermore, the seasonal cycle of the sea ice extent is characterized by many other elements in addition to this asymmetry, such as its amplitude or the timing of the maximum retreat. Factors controlling those characteristics also need to be analyzed to quantify how the seasonal cycle of the Antarctic sea ice influences the dynamics of the climate at high southern latitudes. Models still have large biases on those aspects and a better understanding is necessary for model improvement (Downes et al., 2015; Eayrs et al., 2019; Roach et al., 2020; Raphael et al., 2020; Schroeter and Sandery, 2022).



Several studies have addressed the role of sea ice processes and atmosphere and ocean feedbacks
on Antarctic sea ice extent, focusing both on the mean seasonal cycle and the interannual variability
(e.g., Fichefet and Morales Maqueda, 1997; Holland and Kimura, 2016; Hobbs et al., 2016; Kusahara et
al., 2019). An instructive diagnostic is to decompose the contribution of the dynamics, including the
transport of sea ice, from the one of thermodynamics that influences the local formation or melting of
sea ice. This decomposition is not always straightforward, as for example winds control both the sea
ice transport and the advection of warm or cold air masses that impacts thermodynamic processes.
The results may also depend on the definition of the dynamics and thermodynamics contributions.
Nevertheless, a common conclusion is that the thermodynamic processes play a strong role nearly all
year long, with a clearly dominant contribution during the advance period, while the impact of the
winds becomes more important later in the season, in particular during the retreat (Fichefet and
Morales Maqueda, 1997; Holland and Kimura, 2016; Kusahara et al., 2019; Eayrs et al., 2020).
Despite those advances, many uncertainties remain around the processes controlling the seasonal
cycle of the Antarctic sea ice, in particular because the majority of existing studies address only some
of the processes, forbidding a comparison between different factors, or are devoted to the variability
and trends, not to the seasonal cycle itself. As a consequence, our goal here is to propose an analysis
of the different processes in a single framework, using sensitivity experiments designed to the study
of the seasonal cycle. Specifically, we perform sensitivity experiments with a sea-ice-ocean model
driven by an atmospheric reanalysis and the same model coupled to a regional atmospheric model,
disabling or strongly perturbing key processes related to sea ice dynamics and thermodynamics as well
as the exchanges between the atmosphere and ocean.
The goal of those sensitivity experiments is not to impose realistic changes or to improve
agreement with observations but rather to determine the role of the associated processes. In contrast
to many existing sensitivity studies performed with sea ice-ocean models, the experiments with the
coupled model will address the limitations associated with a prescribed atmospheric state, which tends
to damp the changes imposed by the perturbation as the location of the sea ice edge is strongly
controlled by the atmospheric forcing, in particular in winter (e.g., Urrego-Blanco et al, 2016).
Furthermore, the comparison between the experiments with and without coupling with the
atmosphere will, for the first time, quantify the regional atmospheric feedbacks in response to the
imposed perturbation. The sensitivity experiments last only two years and are not analyzed at
equilibrium for two reasons. First, the drift of the model state after several years in response to the
perturbation can be large. The relative importance of the various processes, which may depend of the
mean state, can thus be very different from the one in the current climate. Second, by comparing the
first years of each experiment, which start with identical conditions at the beginning of the season,
and the second year, for which the perturbation has already acted during one year, we can determine
the contribution of the initial state and the one of the processes occurring during the sea ice advance
and retreat seasons. This approach is also instructive for understanding observed changes and for
predictions as this distinction between initial conditions and ongoing perturbations is key in
interpreting the observed variability. Many studies have demonstrated that large spatial variations are
present between the different sectors of the Southern Ocean (e.g., Parkinson et al., 2019; Kusahara et
al., 2019; Kacimi and Kwok, 2020). Analyzing them is necessary to have a full picture of the dynamics
of the system. Nevertheless, we will focus here first on the ice extent integrated over the whole
Southern Ocean, keeping the regional changes for future work except when critically needed to
interpret the integrated changes. The models used and the perturbation applied are described in
Section 2. Section 3 presents the main results of the sensitivity experiments. Section 4 is devoted to
the atmospheric feedbacks. Section 5 includes a discussion and a synthesis of our main results.






## 2    Methodology

Model description

The simulations are performed with a regional circum-Antarctic configuration of the sea-ice-ocean
model NEMO-LIM3 version 3.6 (Rousset et al., 2015) driven by the ERA5 atmospheric reanalysis
(Hersbach et al., 2020) and with NEMO-LIM3 coupled to the COSMO-CLM$^2$ regional atmospheric model
(Pelletier et al., 2022a). The model set-up and forcing are identical to Verfaillie et al. (2022) for NEMO-
LIM3 driven by ERA5 and to Pelletier et al. (2022a) for NEMO-LIM-COSMO-CLM$^2$, except that, for the
latter, a bug in the interpolation of the winds in the coupling between the ocean and atmosphere has
been corrected (Pelletier et al., 2022b). The version of NEMO-LIM3 driven by ERA5 will hereafter be
referred to NEMO and the version coupled to COSMO-CLM$^2$ as PARASO following Pelletier et al.
(2022a).
NEMO (Nucleus for European Modelling of the Ocean, Madec et al., 2017) includes the OPA ocean
model (Océan PArallélisé) coupled with the Louvain-la-Neuve sea ice model (Vancoppenolle et al.,
2012; Rousset et al., 2015). Our configuration has an explicit representation of Antarctic ice shelf
cavities using the implementation of Mathiot et al. (2017). The free-surface oceanic component is
hydrostatic and applies finite differences to solve the equations on an Arakawa C-grid. Vertical mixing
is computed using a turbulent kinetic energy (TKE) scheme (Gaspar et al., 1990), while lateral diffusion
of momentum is carried out with a bi-Laplacian viscosity and isopycnal diffusion of tracers with a
Laplacian operator. Oceanic convection is represented using an enhanced vertical diffusivity, triggered
under unstable vertical stratification (Lazar et al., 1999). The sea ice component uses an elastic-viscous-
plastic rheology (Bouillon et al., 2013) and a five-category ice-thickness distribution (Bitz et al., 2001;
Massonnet et al., 2019). Each of those categories is covered by snow, with one snow thickness per
category. The energy conserving sea ice thermodynamics follows Bitz and Lipscomb (1999) and
includes an explicit representation of the evolution of salt content and its impact on the sea ice
properties (Vancoppenolle et al., 2009). The albedo of sea ice depends on snow and ice thickness,
surface temperature and cloud cover (Grenfell and Perovich, 2004; Brandt et al., 2005).
The model grid is ePERIANT025 (Mathiot et al., 2017) that has a nominal horizontal resolution of ¼
of a degree with an isotropic spacing, meaning that the resolution is about 24 km at 30°S but increases
up to 3.8 km over the Antarctic continental shelf. A z-coordinate is applied on the vertical using 75
levels, with a thickness of about 1m at surface reaching 200m at depth and partial steps in the bottom
layer (and in the top layer beneath ice shelves). In the uncoupled simulations, NEMO is driven at the
surface by the fluxes computed by the CORE bulk formulas (Large et al., 2004) using 3-hourly fields
derived from the ERA5 reanalysis (Hersbach et al., 2020). The conditions at the northern boundary of
the domain (30°S) are prescribed from the ORAS5 ocean reanalysis (Zuo et al., 2019).
In PARASO, NEMO is coupled to COSMO-CLM$^2$, which includes the version 5.0 of the COnsortium
for Small-scale MOdeling (COSMO) regional atmospheric model (Rockel et al., 2008) and the
Community Land Model version 4.5 (Oleson et al., 2013). COSMO is a non-hydrostatic model using
generalized terrain-following height coordinates with 60 levels (Doms and Baldauf, 2018). The version
utilized here includes parameter calibration adapted to polar regions and a new snow scheme
(Souverijns et al., 2018). Furthermore, the computation of the fluxes is separated over land, ocean and
sea ice surfaces for the coupling with NEMO (Pelletier et al., 2022a). The conditions at the lateral
boundary of the domain are obtained from ERA5. COSMO-CLM$^2$ uses a rotated latitude-longitude grid
with a horizontal resolution of 0.22°, which corresponds to about 25 km. The domain is smaller than





the one of NEMO, with a northern boundary located between 50°S and 40°S. In the areas not simulated
by COSMO-CLM², NEMO is forced by ERA5 fields as in the uncoupled configuration.

190        Experimental design

191        NEMO is driven by the ERA5 reanalysis using every year the forcing from the period 1st May 1990
to 30th April 1991, which is considered a normal period regarding the major modes of climate
variability (Stewart et al., 2020; Verfaillie et al., 2022). This simulation thus has no interannual
variability in order to focus specifically on the seasonal cycle, while keeping conditions close to the
model climatology. The two-year simulations analyzed here follow a 10-year spin-up, which is sufficient
to have a quasi-equilibrium for the surface variables (Verfaillie et al., 2022). The PARASO simulation
has been initialized in 1985 and we discuss here two-year simulations following a 10-year spinup,
meaning that the analyses start in 1995. The conditions are thus slightly different in the two
configurations. Nevertheless, the mean states of the coupled and uncoupled models are also different,
forbidding a direct comparison between them anyway (Fig. 1). Both configurations underestimate the
sea ice extent in summer, and tend to overestimate it in winter. They also have a delayed and too rapid
retreat season. Those biases are similar to those found in many other coupled and uncoupled models
(Downes et al., 2015; Eayrs et al., 2019; Roach et al., 2020; Raphael et al., 2020; Schroeter and Sandery,
2022). Each sensitivity experiment will be compared to the reference simulation using the same model
configuration and initial state, assuming that the biases are small enough to have only a marginal effect
on the response to the perturbation. Additionally, in contrast to NEMO alone, PARASO can develop
some internal variability despite the strong constraints at its boundaries. Ideally, an ensemble of
simulations should be performed for each of the coupled experiments, but this exceeds available
computing capacities. Tests with identical configurations but slight perturbations of the initial state
indicate that the difference in ice extent is usually much smaller than 0.2 million km², i.e. less than the
response to the perturbation in the majority of the experiments, but the possibility that some of the
differences between the experiments are just occurring by chance must be kept in mind.

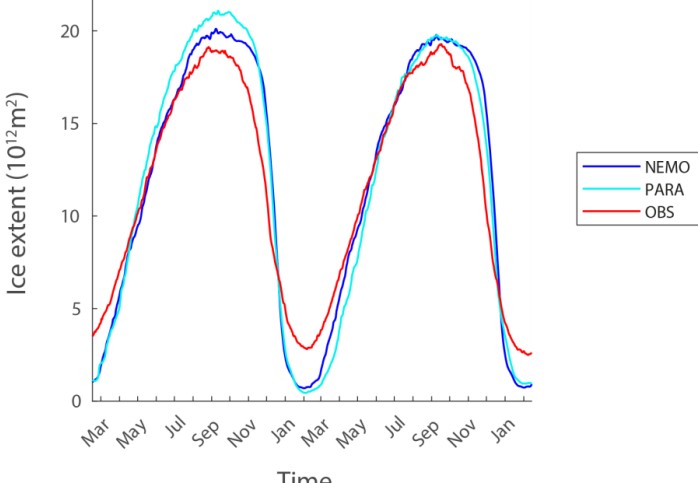


Figure 1. Seasonal cycle of the Antarctic sea ice extent (in $10^{12}$ m²) in observations (Fetterer et al.,
2017) and in the reference experiments with NEMO and PARASO (starting in March). For observations
and PARASO, the period from March 1995 to February 1997 is shown while for NEMO the forcing
corresponds to the 'normal period' from May 1990 to April 1991 that is applied twice.



Set-up of the sensitivity experiments
Identical perturbations are applied in NEMO and PARASO on the 1st of March and 1st of September
in the 2-year sensitivity experiments (see Table 1). The first two experiments are devoted to the role
of the exchanges between sea ice and the oceanic mixed layer. In the first one (Mix100), the ocean
temperatures and salinities are homogenized from the surface to 100m depth at each time step by
completely mixing the corresponding grid boxes in each column. This depth roughly corresponds to
the seasonal maximum depth of the mixed layer in the model in most ice-covered regions except over
the continental shelf (e.g., Barthélemy et al., 2015). The effect of this mixing scheme perturbation is
that the seasonal summer shoaling of the mixed layer due to freshening is removed. The goal is to
determine whether such deep summer mixing favors heat storage at the surface and delays the sea
ice advance. In the second experiment (NoMassFlux), sea ice growth and melt is no longer associated
with freshwater uptake and release. This is equivalent to assuming that sea ice salinity is the same as
the ocean surface salinity. Therefore, the surface ocean salinity no longer responds to sea ice formation
and melting. This disables the negative ice production-entrainment feedback (Martinson, 1990) in
which the upper ocean salinity increase due to ice formation induces a mixed layer deepening and
entrainment of deeper warmer water towards the surface that reduces ice formation. The absence of
this negative feedback in NoMassFlux could thus potentially accelerate the sea ice advance.
The second group of experiments is devoted to sea ice physics and properties. As sea ice thickness
is a key characteristic of the pack that strongly controls its behavior, the first two experiments
artificially increase (ThickIce) and decrease (ThinIce) the ice thickness. This is achieved by increasing
(ThickIce) and decreasing (ThinIce) the thermal conductivities of the ice and snow by a factor of five.
With low conductivity, ice becomes a much better insulator for the ocean that loses less heat to the
atmosphere in fall and winter, inducing a slower increase in ice thickness. We expect then that a
thinner ice will melt faster in spring, accelerating the ice retreat. As the ice-albedo feedback is expected
to be a dominant element of the seasonal sea ice retreat, setting both the albedo of the snow and ice
to the ocean value in AlbOce should accelerate the retreat.
We also test the impact of ice dynamics by disabling it (NoIceDyn). The ice dynamics are expected
to favor a faster sea ice advance in fall by transporting sea ice from the colder regions, where it is
quickly replaced because of strong ice formation, to the north where it can survive because of the
relatively low temperature. It also accelerates the retreat in spring by transferring sea ice to regions
where it is warm enough during this season to quickly melt and by creating leads within the pack that
enhances the ice-albedo feedback. Suppressing ice dynamics should thus reduce the amplitude of the
seasonal cycle of the sea ice extent. For technical reasons, the implementation of sea-ice dynamics
suppression differs in uncoupled and coupled experiments: in the former, all the sea-ice dynamic
components of the model are disabled; in the latter, solely the velocity and large-scale transport is set
to zero in PARASO (other mechanisms such as ridging are active).
Although no sensitivity experiment includes explicit modifications of atmospheric parameters or
processes, all of the applied perturbations affect indirectly the exchanges between the ocean-sea-ice
system and the atmosphere by modifying the surface conditions. Comparing the coupled and
uncoupled configurations quantifies the contribution of the atmospheric feedbacks.





Table 1. List of experiments. Each experiment is performed for 2 years with NEMO and PARASO
and for two starting dates, March 1 and September 1. For references in the text, NEMO and PARASO
experiments have the additional suffixes NEMO and PARA, respectively, while for the two starting
dates we use the suffixes Mar and Sep.

| Short name | Description |
|---|---|
| Ref | Reference experiment without perturbation |
| Mix100 | Ocean mixed over the top 100m of the ocean all year long |
| NoMassFlux | No mass flux associated with the sea ice formation or melting |
| ThickIce | Sea ice and snow thermal conductivities divided by 5 |
| Thinice | Sea ice and snow thermal conductivities multiplied by 5 |
| AlbOce | Sea ice and snow albedos equal to that of the ocean (=0.088) |
| NoIceDyn | Ice dynamics disabled (uncoupled mode); or sea ice velocity equals zero (coupled mode). |


## 3   Results

*First advance season*
In the sensitivity experiments starting in March, the perturbations applied to the model physics
have very little impact on the sea ice advance until August (Fig. 2ab), both in the coupled and
uncoupled model configurations. When starting from identical initial conditions, the sea ice advance
seems thus controlled by external conditions imposed by the seasonal evolution of the insolation and
does not depend much on the sea ice physics or on the interactions between sea ice, the ocean and
the atmosphere. Even the absence of sea ice transport (experiment NoIceDyn_NEMO_Mar and
NoIceDyn_PARA_Mar) has nearly no effect on the total sea ice extent during this period, confirming
previous studies indicating that the sea advance is mainly of thermodynamic nature (e.g., Fichefet and
Morales Maqueda, 1997; Kusahara et al., 2019). The impact on the sea ice volume is more immediate,
with a difference that can reach more than a factor two in August between some experiments such as
ThickIce_NEMO_Mar and Thin_NEMO_Mar (Fig. 3a). Nevertheless, this change in volume has little
impact on the extent, showing a decoupling between the two variables in our experiments during this
first advance season.
The different experiments have varying ice growth rates, consistent with the differences in ice
volume, but the temporal evolution is relatively similar during the advance season (Fig. 4). ThickIce
and NoMassflux stand as exceptions. In ThickIce, the ice production-entrainment feedback is very
active as a consequence of the large sea ice formation and brine release that destabilizes the water
column. The oceanic mixed layer depth (Fig. S1) is thus much larger than in the other experiments and
the associated vertical ocean-to-ice sensible heat transfer compensates early in the season for a
significant fraction of the cooling imposed at surface, explaining the early peak in the freezing rate (for
instance the peak occurs in day 166 in ThickIce_NEMO_Mar compared to day 220 in the corresponding
reference simulation). In NoMassFlux, by contrast, as the ice production-entrainment feedback is
inactive by design, the oceanic mixing is much weaker and strong ice formation can be sustained until
the end of the growth season, particularly in the PARASO configuration, with a peak in ice formation
in NoMassFlux_PARA_Mar on day 247 compared to day 187 in the corresponding reference simulation.
*Maximum extent and retreat season*
The modification of the ice volume imposed by the perturbations has only a weak impact on the
sea ice extent until August, as indicated above, but experiments with thicker ice tend to have a larger



sea ice extent after August, a longer plateau with an extent close to the maximum, and a slower retreat.
For instance, ThickIce_NEMO_Mar, which has the largest volume for all the experiments with NEMO,
has a maximum ice extent that is higher than in Thin_NEMO_Mar by 1.2 million $km^2$, a delayed
beginning of the retreat in this experiment, and an extent that is larger than in Thin_NEMO_Mar by
3.3 million $km^2$ at the end of November (Fig. 2a). The impact of volume differences on the date of the
maximum extent itself is generally weak (see Tables 2 and 3), but a link between the maximum volume
and the date at which the sea ice extent decreases to 95% of its maximum is clear in most experiments
(Fig. 5a).
Thicker ice in September tends thus to delay sea ice retreat, as expected. However, the conditions
in September (which integrate the effect of the perturbation in model physics since March in the
simulations started at that time) are not the only reason for the difference between the experiments
during the retreat season. The experiments starting in September from identical initial conditions tend
to diverge nearly immediately, indicating a larger control of sea ice physics on the evolution of the ice
extent at this time of the year compared to the advance season (Fig. 2cd).
This large role for sea ice physics in the melt season is illustrated by the larger differences between
the experiments for the timing of maximum of the ice melting than for the timing of maximum ice
growth rate (Fig. 4). The maximum ice melting rate spans a range of up to 50 days between the
experiments that have the earliest melting (AlbOce) and the latest one (ThickIce and No_Ice_Dyn). The
faster and earlier melting occurs in experiments AlbOce, as the low albedo in those experiments allows
a stronger absorption of incoming solar radiation and thus a larger amount of melt as soon as the Sun
is high enough above the horizon. In AlbOce experiments, a large part of the retreat is already achieved
by the end of November. This leads to a difference in ice extent that can reach more than 10 million
$km^2$ compared to the reference experiments at this time and thus a sea ice extent corresponding to
the one simulated only in early January in these reference experiments (Fig. 2). The ThinIce
experiments also display an earlier melting than ThickIce ones, reinforcing the direct effect of the
initially thinner ice in winter. This is due to a more efficient ice-albedo feedback: it is easier to melt
thin sea ice, leading to a higher amount of open water and thus a larger absorption of incoming solar
radiation and an intensified melting.
*Minimum extent, subsequent advance season and amplitude of the seasonal cycle*
Experiments with earlier melt onset and larger melt rates show faster retreat and lower minimum
extent, leading to a larger difference between the experiments in the first summer than in the first
winter. In the experiments starting in March, the range of ice extent across all experiments at the first
maximum reaches 1.2 million $km^2$ for NEMO and 1.9 million $km^2$ for PARASO. For the following
minimum in the same experiments, it reaches 3.6 million $km^2$ and 3.8 million $km^2$, respectively. The
numbers for the summer minimum are relatively similar for the experiments starting in September
compared to those starting in March, which suggests that processes in the summer season are more
important than the state of the sea ice-ocean-atmosphere system in September (Tables 2 and 3).
By contrast, the state of the sea ice-ocean-atmosphere system in March (i.e. the second year for
the experiments starting in March but already the first year for the experiments starting in September)
has a dominant influence during the whole sea ice advance season (Marchi et al., 2020). Despite the
strong control from the insolation and the limited direct impact of sea ice physics and feedbacks with
the ocean and the atmosphere during the first advance season (see above), the model physics
influences thus the evolution of the sea ice extent for several months during the second advance
season through their effect on the state of the system in March. This is illustrated in Fig. 5b by the
association between positive minimum sea ice extent anomaly and the subsequent positive maximum





extent anomalies present in most experiments, with the notable exception of NoIceDyn experiments
as discussed below. In this figure, the minimum sea ice extent is chosen as a proxy for the state of the
sea ice and ocean system in summer but a similar link can be found for other variables, such as the
mean summer sea surface temperature southward of 60°S (Fig. S2).
The role of the state of the system in March can be illustrated for example using the Mix100
experiments. Increasing the vertical oceanic mixing in the sensitivity experiments redistributes the
available energy over the top 100 meters without modifying the vertically integrated heat content.
This does not have a large influence initially in the experiments starting in March (Fig. 2a). However,
the second year in the experiments starting in March is different from the first year as a deeper mixed
layer allows a larger heat uptake in summer. Consequently, the Mix100 experiments tend to have a
smaller ice extent than the reference experiments during the second sea ice advance season (Fig. 2a).
More generally, for both the coupled and uncoupled experiments, the summer extent influences
the whole advance season and the maximum extent. However, the difference in sea ice extent
between the experiments with NEMO tends to decrease with time because of the restoring imposed
by a fixed atmospheric state. For instance, the range in the maximum extent for the second year of the
experiments beginning in March reaches 2.1 million km$^2$ while it was 3.6 million km$^2$ the previous
summer (Fig. 2a). By contrast, the range between experiments increases during the sea ice advance
season in the PARASO experiments, reaching 4.8 million km$^2$ for the maximum extent in winter (25%
more than for the summer minimum).
While the majority of the experiments displaying a large winter ice extent also have a larger
summer ice extent, inducing relatively modest changes in the amplitude of the seasonal cycle, this is
not the case in the NoIceDyn experiments. Those experiments are characterized by a reduced
amplitude of the seasonal cycle of the sea ice extent, with a smaller extent in winter and a larger one
in summer compared to the reference experiments. At the end of the advance season, the ice edge
position is set by the advection of sea ice from the south. Sea ice then melts in regions which are too
warm to sustain local production (e.g., Holland and Kimura, 2016; Nie et al., 2022). Neglecting ice
transport thus leads to an earlier maximum extent and onset of the retreat (Fig. 2). Later during the
retreat season, ice is transported northward where it melts and this transport also enhances the
formation of leads within the ice pack that increases solar absorption. Therefore, ice dynamics plays
an important role in accelerating the ice retreat, as shown in earlier studies (Fichefet and Morales
Maqueda, 1997; Holland and Kimura, 2016, Kusahara et al., 2019; Eayrs et al., 2020), and neglecting
this effect in NoIceDyn induces an increase in the minimum ice extent of several million km$^2$ (Tables 2
and 3).
*Sensitivity to the starting date in* NoMassFlux
Neglecting brine release during ice formation (experiments NoMassFlux) reduces the heat input
from the deeper oceanic layer to the surface and results in a clear increase in ice production and
volume in the experiments started in March, in particular in coupled mode. It only has a limited
influence on the sea ice extent during the first winter as, by definition, it can only act after sea ice is
already present (Martinson, 1990). The effect can only be seen indirectly during the sea ice retreat
season (when entrainment no longer plays a clear direct role) and the second year, through the
influence of the perturbation on the sea ice volume. In particular, this leads to an increase in sea ice
extent in NoMassFlux_PARA_Mar of nearly 2.0 million km$^2$ compared with the corresponding
reference experiment in summer.
The NoMassFlux experiments starting in September have a different behavior than the
experiments beginning in March. As the model has a very low sea ice volume in March, assuming that





sea ice has the same salinity as the ocean does not substantially impact the salt and freshwater balance
of the model. In contrast, for the experiments starting in September, because of the much larger initial
sea ice volume, the NoMassFlux experiments imply a large artificial salt input in the system. The salt
input weakens the upper ocean stratification, enhances mixing and triggers open ocean convection
and the formation of open ocean polynyas (Fig. 6). This brings a large amount of heat to the ocean
surface, reducing both the sea ice volume and winter sea ice extent in NoMassFlux_NEMO_Sep and
NoMassFlux_Par_Sep compared with the corresponding reference experiments.
*Timing of the maximum and minimum extents*
Overall, as expected based on previous studies, the effect of the perturbations prescribed in our
sensitivity experiments is relatively modest on the timing of the minimum and maximum ice extents.
The largest signal arises in the sea ice dynamics perturbation, which tends to advance the date of
maximum in the coupled experiments (14 days and 12 days for the second maximum in
NoIceDyn_PAR_Mar and NoIceDyn_PAR_Sep, respectively), and in the experiment with perturbed
heat conduction, as the thicker pack can delay the maximum by up to 25 days (in ThickIce_NEMO_Sep).
Open ocean convection can also bring forward the date of the maximum with a third maximum already
achieved in day 230 and day 239 the second year in NoMassFlux_NEMO_Sep and
NoMassFlux_PAR_Sep (36 and 28 days earlier compared to the previous year of the same experiment,
respectively). The summer minimum can be advanced by up to 43 days in AlbOce_PAR_Sep through
the albedo decrease, and delayed by up to 18 days in the sea-ice dynamics deprived experiment
NoIceDyn_NEMO_Sep. Note that some values in Tables 2 and 3 should be taken with caution as the
evolution of sea ice extent is relatively flat close to the maximum and small differences can produce
large shifts in the specific day of the maximum (e.g. in Mix100_PAR_Sep and ThinIce_PAR_Sep).



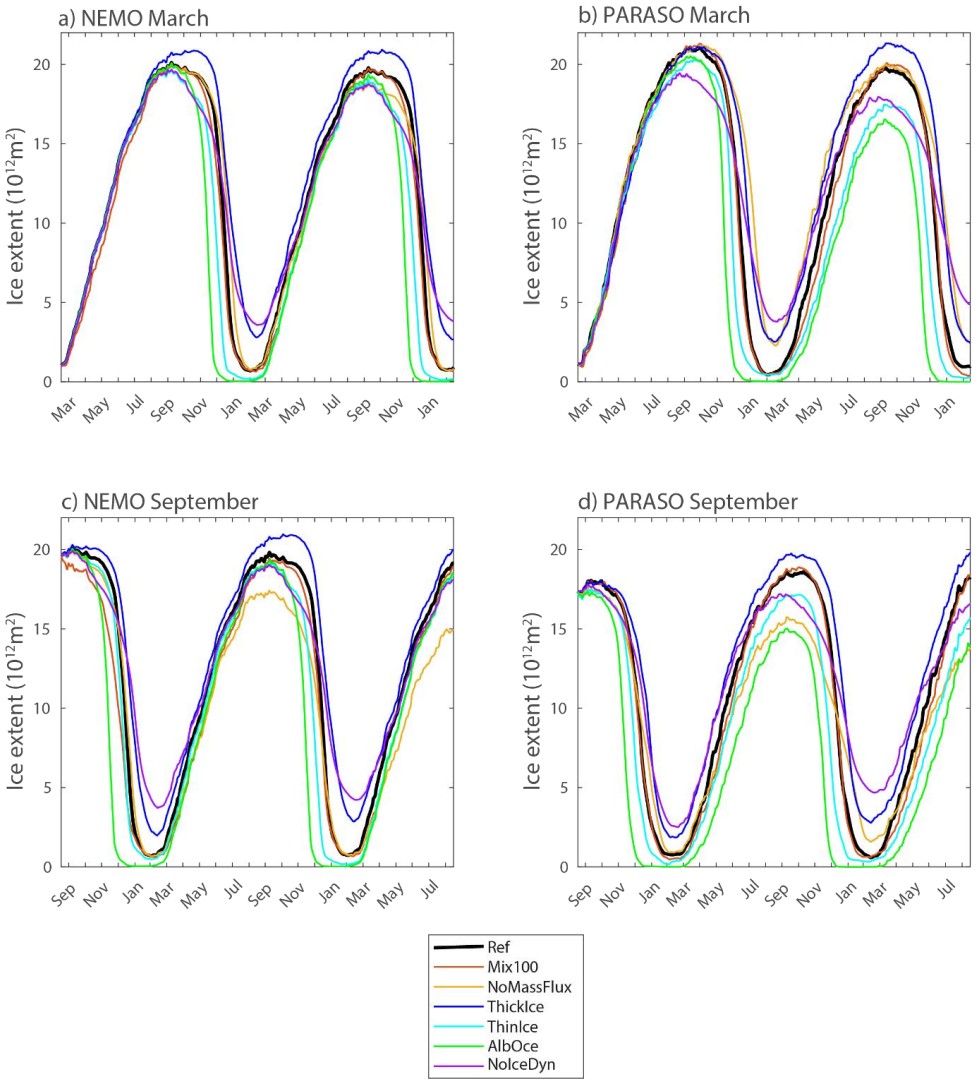

Figure 2. Antarctic sea ice extent (in $10^{12}$ m$^2$) in the group of experiments starting in March
(top row) and September (bottom) for the NEMO (left column) and PARASO configurations
(right column).

Figure 3. Antarctic sea ice volume (in $10^{12}$ m$^3$) in the group of experiments starting in March
(top row) and September (bottom) for the NEMO (left column) and PARASO configurations
(right column).





Figure 4. Mass flux due to sea ice growth and melt (counted positive for melting) integrated over the Southern Ocean (in $10^{11}$ m$^3$d$^{-1}$) in the group of experiments starting in March (top row) and September (bottom) for the NEMO (left column) and PARASO configurations (right column). This diagnostic in evaluated online in NEMO from the different contributions to ice formation and melting but is equivalent to the time derivative of the ice volume.



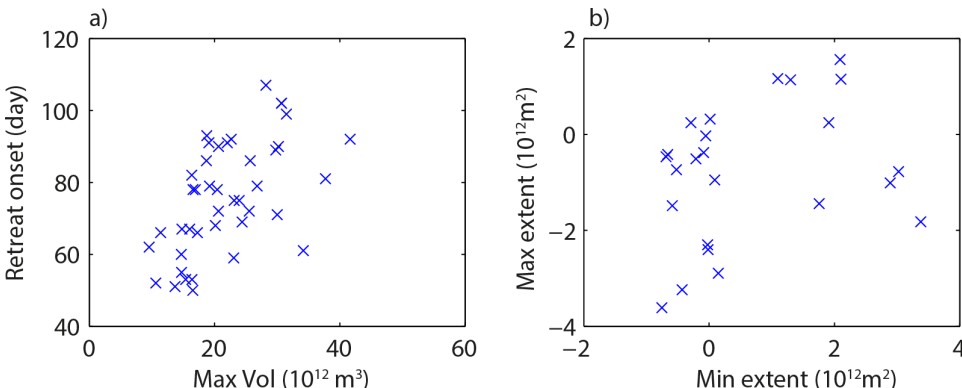


Figure 5. a) Onset of significant seasonal sea ice retreat (in day), defined as the number of days
after the maximum at which the Antarctic sea ice extent has decreased to 95% of its maximum value
as a function of the maximum ice volume (in $10^{12}m^3$). b) Maximum sea ice extent anomaly (in $10^{12}m^2$)
compared to the reference experiment as a function of the anomaly in the previous minimum (in
$10^{12}m^2$) for the second year of the experiments starting in March and for the first minimum and second
maximum of the experiments starting in September.

Table 2. Values and timings of the maximum and minimum sea ice extents for the two years of the
sensitivity experiments starting in March. Extents are given in $10^{12}$ $m^2$ and timings in Julian days.

|  | Year 1 |  |  |  | Year 2 |  |  |  |
|---|---|---|---|---|---|---|---|---|
|  | Max | Min | Day Max | Day Min | Max | Min | Day Max | Day Min |
| Ref_NEMO_Mar | 20.1 | 0.70 | 265 | 46 | 19.8 | 0.73 | 266 | 47 |
| Mix100_NEMO_Mar | 20.0 | 0.64 | 265 | 57 | 19.8 | 0.66 | 266 | 57 |
| NoMassFlux_NEMO_Mar | 20.0. | 0.79 | 265 | 46 | 18.8 | 0.70 | 266 | 49 |
| ThickIce_NEMO_Mar | 20.9 | 2.80 | 265 | 58 | 20.9 | 2.66 | 291 | 58 |
| Thinice_NEMO_Mar | 19.7 | 0.17 | 265 | 48 | 19.0 | 0.12 | 266 | 47 |
| NoIceDyn_NEMO_Mar | 19.7 | 3.59 | 265 | 59 | 18.8 | 3.58 | 266 | 60 |
| AlbOce_NEMO_Mar | 20.1 | 0.03 | 265 | 43 | 19.4 | 0.01 | 266 | 43 |
| Ref_PAR_Mar | 21.1 | 0.45 | 269 | 48 | 19.8 | 0.59 | 267 | 60 |
| Mix100_PARA_Mar | 21.3 | 0.46 | 286 | 48 | 20.1 | 0.37 | 269 | 57 |
| NoMassFlux_PARA_Mar | 21.2 | 2.36 | 294 | 59 | 20.0 | 2.29 | 267 | 63 |
| ThickIce_PARA_Mar | 21.1 | 2.54 | 294 | 59 | 21.3 | 2.45 | 267 | 59 |
| Thinice_PARA_Mar | 20.3 | 0.42 | 276 | 48 | 17.5 | 0.23 | 264 | 50 |
| NoIceDyn_PARA_Mar | 19.4 | 3.82 | 249 | 59 | 17.9 | 3.79 | 253 | 65 |
| AlbOce_PARA_Mar | 20.5 | 0.02 | 269 | 57 | 16.5 | 0.00 | 265 | 30 |







Table 3. Values and timings of the maximum and minimum sea ice extents for the two years of the sensitivity experiments starting in September. Extents are given in $10^{12}$ m$^2$ and timings in Julian days.

|  | Year 1 |  |  |  | Year 2 |  |  |  |
|---|---|---|---|---|---|---|---|---|
|  | Max | Min | Day Max | Day Min | Max | Min | Day Max | Day Min |
| Ref_NEMO_Sep | 20.1 | 0.69 | 265 | 44 | 19.8 | 0.73 | 266 | 45 |
| Mix100_NEMO_Sep | 19.6 | 0.60 | 244 | 57 | 19.4 | 0.66 | 266 | 55 |
| NoMassFlux_NEMO_Sep | 19.9 | 0.67 | 265 | 49 | 17.4 | 0.69 | 266 | 47 |
| ThickIce_NEMO_Sep | 20.3 | 1.99 | 265 | 57 | 20.9 | 2.66 | 286 | 56 |
| Thinice_NEMO_Sep | 20.0 | 0.48 | 265 | 44 | 19.3 | 0.12 | 266 | 44 |
| NoIceDyn_NEMO_Sep | 20.0 | 3.71 | 265 | 59 | 19.0 | 3.58 | 266 | 63 |
| AlbOce_NEMO_Sep | 20.0 | 0.03 | 265 | 44 | 19.3 | 0.01 | 266 | 41 |
| Ref_PARA_Sep | 18.0 | 0.76 | 268 | 56 | 18.6 | 0.58 | 267 | 58 |
| Mix100_PARA_Sep | 18.0 | 0.46 | 269 | 50 | 18.9 | 0.61 | 290 | 56 |
| NoMassFlux_PARA_Sep | 17.7 | 0.90 | 262 | 57 | 15.7 | 1.58 | 267 | 58 |
| ThickIce_PARA_Sep | 18.0 | 1.85 | 287 | 61 | 19.8 | 2.78 | 275 | 58 |
| Thinice_PARA_Sep | 17.5 | 0.16 | 269 | 47 | 17.2 | 0.32 | 289 | 54 |
| NoIceDyn_PARA_Sep | 17.8 | 2.52 | 262 | 63 | 17.2 | 4.68 | 255 | 65 |
| AlbOce_PARA_Sep | 17.4 | 0.00 | 244 | 64 | 15.0 | 0.00 | 267 | 15 |


a)            b)

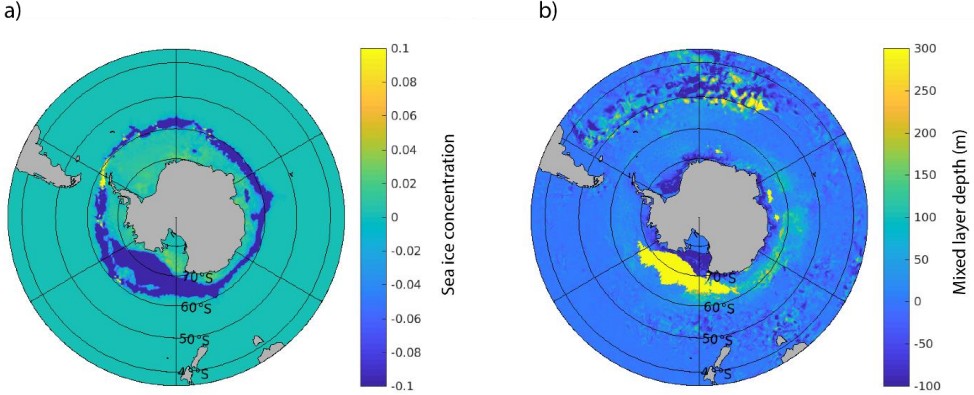


Figure 6. Differences in a) ice concentration and b) mixed layer depth (in m) in August of the second year of simulation between NoMassFlux_NEMO_Sep and the corresponding reference experiment.


## 4 Atmospheric feedbacks

The results discussed in Section 3 have highlighted differences between the NEMO and PARASO experiments and the role of the coupling with the atmosphere is further quantified here. In NEMO, the surface energy budget has only one degree of freedom (the surface temperature). Therefore, the surface readily adjusts to the forcing, so that the surface temperature closely follows the air temperature, which can be seen as a form of restoring. In PARASO, the surface energy budget responds to both sea ice and atmospheric processes. Another degree of freedom is now that the atmosphere





warms or cools in response to changes in sea ice, which in turn affects non-solar (downward longwave
and turbulent) fluxes.
This effect of the changes in the atmosphere is evaluated by computing atmospheric feedback
factors in response to the perturbation for each pair of coupled and uncoupled model experiments.
Feedbacks can be evaluated in different ways. A methodology that is consistent for a wide range of
feedbacks, including the standard radiative ones involved in computation of the so-called climate
sensitivity as well as non-radiative feedbacks, is to define the feedback factor $\gamma$ as (Goosse et al., 2018):
$$\gamma = \frac{Total\ Response - Reference\ Response}{Total\ Response} \qquad (1)$$

where the *Total Response* corresponds to the response of the model to some perturbation imposed in
the system when all the feedbacks are active, while the *Reference Response* is the response of the
model to the same imposed perturbation when one feedback or process to be studied (for instance
sea ice dynamics) has been left out. As our specific goal is to study the impact of atmospheric coupling,
this leads to:
$$\gamma = \frac{Coupled\ Response - Uncoupled\ Response}{Coupled\ Response} \qquad (2)$$

and for sea ice extent specifically:
$$\gamma_{SIE} = \frac{\Delta SIE_{PARA} - \Delta SIE_{NEMO}}{\Delta SIE_{PARA}} \qquad (3)$$

where $\Delta SIE_{PARA}$ and $\Delta SIE_{NEMO}$ are the changes between the sensitivity experiments and the reference
experiments in the PARASO and NEMO configurations, respectively.
The feedback factor can be related to the feedback gain *G* (e.g., Goosse et al. 2018) defined here
as the ratio between the response in coupled mode and the one in uncoupled mode:
$$G = \frac{\Delta SIE_{PARA}}{\Delta SIE_{NEMO}} = \frac{1}{1 - \gamma_{SIE}} \qquad (4)$$

A negative value of $\gamma$ thus corresponds to a negative feedback (changes in PARASO smaller than in
NEMO, feedback gain smaller than 1, and the feedback dampens the response to a perturbation); a
value between 0 and 1 corresponds to a positive feedback (changes in PARASO larger than in NEMO,
feedback gain larger than 1, the feedback amplifies the perturbation); a value of 1 implies an infinite
gain and values of $\gamma$ larger than 1 imply a change in the sign of the response between coupled and
uncoupled model experiments (negative feedback gain). In the following, we start by discussing the
feedback factors lower than 1 (positive and negative feedbacks and positive feedback gains) that are
the easiest to interpret in a linear framework, while non-linearities and values of $\gamma$ larger than one
(negative feedback gain) will be discussed in the last paragraphs of the section. We have not analyzed
the feedback factors when the coupled response is smaller than 0.2 million km$^2$ for sea ice extent or
0.2 thousand km$^3$ for sea ice volume, corresponding to very small changes in the system and large
feedback factors (the coupled response appears in the denominator of $\gamma$). Consistently, we have
focused the analyses on the second year of the experiments, as for the first year the changes in several
experiments are too small.



*Atmospheric feedbacks on the maximum ice extent.*
The feedback factors are always positive for the maximum sea ice extent (Fig. 7a), indicating that
the coupling with the atmosphere amplifies the wintertime response to perturbations (for the
feedback factors smaller than 1, for the ones larger than 1 see below). This matches well our
understanding of the system, where sea ice acts as an insulator between the atmosphere and the
ocean. An increase in sea ice extent resulting from a perturbation thus cools the atmosphere, which
amplifies the initial change, giving a positive feedback. The same positive feedback mechanism applies
in the context of an initial decrease in ice extent, leading to atmospheric warming and additional
decrease in extent. For example, in AlbOce_PARA_Mar, the surface air temperature is higher than in
the reference experiment all year long. The difference reaches 1.5K on average over the two years of
the simulations for the oceanic region south of 60°S, and more than 2.5K in the second winter (Fig. 8,
Fig. S3).
Among all the experiments, AlbOce displays the largest feedback gain for the winter ice extent
(i.e. $\gamma$ <1 and closest to 1), with values of $\gamma$=0.87 (Fig. 7a) in both experiments started in March and
September and hence a feedback gain of 7.7 (Fig. S4a). This is not surprising as the albedo changes
associated to sea ice variations are usually considered as a key characteristic of polar marine climates.
The sea-ice albedo feedback is already active in the NEMO configuration as a change in sea-ice
concentration affects the surface albedo and thus the net solar radiation absorbed at surface: in
AlbOce_NEMO_Mar and AlbOce_NEMO_Sep, the ocean-sea ice surface south of 60 S have a net solar
absorption higher than in their reference counterparts of 13 W m$^{-2}$ in annual mean (Fig. S5). This is
even higher than in AlbOce_PARA_Mar and AlbOce_PARA_Sep, where the change reaches only
7 W m$^{-2}$. The higher values in the NEMO configuration might be due to differences in the mean state
between the coupled and uncoupled model configurations or to feedbacks related to clouds in
PARASO, but investigating those effects in more detail is out of the scope of the present study.
Nevertheless, the main difference between the coupled and uncoupled experiments comes from the
non-solar heat fluxes (Fig. S6), which is the net downward flux associated with incoming and outgoing
longwave radiation, and latent and sensible heat exchange with the atmosphere. In
AlbOce_NEMO_Mar and AlbOce_NEMO_Sep, as the atmospheric state is prescribed, the reduction in
sea ice extent and surface warming induce a large increase in non-solar heat losses that reaches 10
and 13 Wm$^{-2}$ averaged over the area south of 60°S, respectively. In other words, the artificial restoring
to the observed atmospheric state in uncoupled mode makes the non-solar heat loss at the surface
nearly compensate for the additional solar heat input. By contrast, the atmospheric warming in
AlbOce_PARA_Mar and AlbOce_PARA_Sep only leads to a moderate increase of the non-solar heat
losses, with annual mean values of 1 and 4 Wm$^{-2}$, respectively. This explains the larger changes in ice
extent in coupled mode and the strong drift of the system to a warmer state (Fig. 8).
*Atmospheric feedbacks on maximum ice volume.*
The feedback factor for the winter volume is also positive in many experiments (Fig. 7b). In
particular, the value of $\gamma$ in NoMassFlux_Mar equals 0.87, corresponding to a feedback gain *G* of 7.7.
In NoMassFlux experiments, the heat input from the ocean to the surface is reduced because of the
absence of the ice production–entrainment feedback. This increases ice production and thus ice
thickness. In the coupled model integration, the downward non-solar (net LW and turbulent) fluxes
can respond to thicker ice and colder surface, which further decreases the surface air temperature by
more than 3K in average over the oceanic region South of 60°S during the sea ice growth season. This
further enhances the ice production and leads then to a very strong positive atmospheric feedback.



By contrast, the atmosphere provides a negative feedback in the case of the ThinIce and ThickIce
experiments. Larger snow and ice thermal conductivities in ThickIce imply larger heat losses from the
ocean to the atmosphere in ice-covered regions and thus larger winter sea ice production in all the
ThickIce experiments (Fig. 4). In the PARASO configuration, the increased heat conduction from the
ice-ocean system warms the lower atmosphere in winter within the ice pack by more than 3K,
integrating over the region south of 60°S (Fig. 8). Consequently, the non-solar atmosphere-ice heat
fluxes can increase in coupled mode, moderating the increase in sea ice volume compared to the
NEMO experiments. In ThinIce, the smaller heat conduction fluxes induce an atmospheric cooling in
winter, located mainly close to the continent where the largest volume change occurs compared to
the reference experiment.
The experimental design in ThickIce and ThinIce may appear counterintuitive as our modifications
to the model physics warm the atmosphere when the ice is thicker. Such perturbations highlight a
coupling between heat conduction in the ice and non-solar downward atmospheric heat fluxes. When
the full system is considered in the real world, we rather experience the effects of the strong coupling
between thickness and heat conduction, often referred to as the ice growth-thickness feedback in
which an anomalously thin sea ice cover will lose more energy by conduction in winter, leading to a
thicker and colder ice, reducing the initial anomaly (Maykut, 1986; Bitz and Roe, 2004; Goosse et al.,
549 2018).

*Atmospheric feedbacks on minimum ice extent and volume.*
Positive feedback factors associated to the coupling with the atmosphere would also be expected
for the minimum ice extent (Fig 7c), in particular because of the amplifying role of the ice-albedo
feedback and its impact on air temperature. This is consistent with the highest summer air
temperature in the two experiments with the lowest summer ice extent (ThinIce and AlbOce, Fig. 8).
Accordingly, positive values are found in several experiments. However, negative values are also
obtained for others. This may be surprising in particular for AlbOce but this can be considered as an
artefact related to the methodology used to compute $\gamma$. All the sea ice melts in summer in the
experiments AlbOce (Figs. 2 and 3). The response is thus equal to the summer sea ice extent (or
volume) in the corresponding reference experiments. As this reference extent (and volume) is slightly
higher in NEMO configuration than in PARASO (Figs. 1 and 2), the response is larger in NEMO. This then
leads to a negative value of $\gamma$ by definition (Eq. 3).
For ThinIce and ThickIce experiments, the negative atmospheric feedback factors obtained for the
summer ice volume (Fig 7d) are a direct consequence of the negative values discussed above for winter
ice volume in the same experiments, the winter sea ice thickness anomalies persisting until the
summer. As those anomalies are particularly large close to the coast, they affect the melting in those
regions and thus the feedback factor for the summer sea ice extent, leading to a negative value in
ThinIce and values very close to zero in the ThickIce experiments (Fig. 7c).
*Feedback factors larger than one: impact of the spatial distribution of the response.*
The analyses of the feedback factors illustrate the nonlinearity of the system, for example when
comparing the very different values of $\gamma$ for an increase or a decrease in the conductivity in ThickIce
and ThinIce. Values of $\gamma$ higher than one also suggest more complex dynamics than a simple
amplification or damping of the response by interactions with the atmosphere as even the sign of the
response is different between coupled and uncoupled model configurations. In many cases, this
different sign of the response integrated over the whole Southern Ocean, as measured on the anomaly
of total sea ice extent or ice volume, is due to a spatially heterogenous response in uncoupled mode.



The coupling amplifies or damps the response locally as described by the feedback framework.
However, this may change the balance between positive and negative contributions and thus modify
the sign of the response integrated over the whole Southern Ocean compared to the uncoupled mode,
explaining the value of $\gamma$ higher than 1.
We will not discuss here all the experiments displaying a value of $\gamma$ higher than 1, especially
because in some cases the difference in the response to the coupling is small and thus probably not
very meaningful. Nevertheless, two examples seem illustrative and are detailed below. In NoIceDyn,
the sea ice thickness increases in winter close to the coast and decreases close to the ice edge
compared to the reference experiment, both in coupled and uncoupled mode (Fig. S7). The integrated
volume response is thus a balance between the changes in the two regions and, depending on their
relative strength, the sign of the change in ice volume can change. In coupled mode, the very large
increase in thickness close to the coast associated with strong local positive feedbacks with the
atmosphere dominates, while in the uncoupled mode, the offshore decrease dominates, then leading
to $\gamma$ greater than 1 for winter ice volume.
At the time of the winter maximum in sea ice extent, sea ice is transported to the ice edge where
it tends to melt. The associated freshwater release increases the upper ocean stratification in the
reference experiment, reducing the oceanic heat input to the surface and thus favoring the advance
of the pack. (This positive feedback at the ice edge at the time of the maximum ice extent can be
contrasted with the negative ice production-entrainment feedback within the pack). In
NoMassFlux_NEMO_Mar, the absence of freshwater release during ice melt leads to a weaker upper
ocean stratification close to the ice edge, allowing deeper mixed layers, with a difference that can
reach more than 100m. As a consequence, the heat input from the ocean to the ice is higher. This is
sufficient to limit the seasonal sea ice advance and the maximum ice extent is lower in
NoMassFlux_NEMO_Mar than in the reference experiment by about 1 million km$^2$ in the second year
of the experiments (Fig. 2a). By contrast, the large increase in ice thickness and volume in
NoMassFlux_PARA_Mar discussed previously dominates the response even at the ice edge, leading to
a positive anomaly in the maximum ice extent. As a consequence, the atmospheric feedback factor is
greater than one. This effect is only seen in the experiments starting in March, as those starting in
September are dominated by the consequences of deep mixing and polynya formation within the pack.

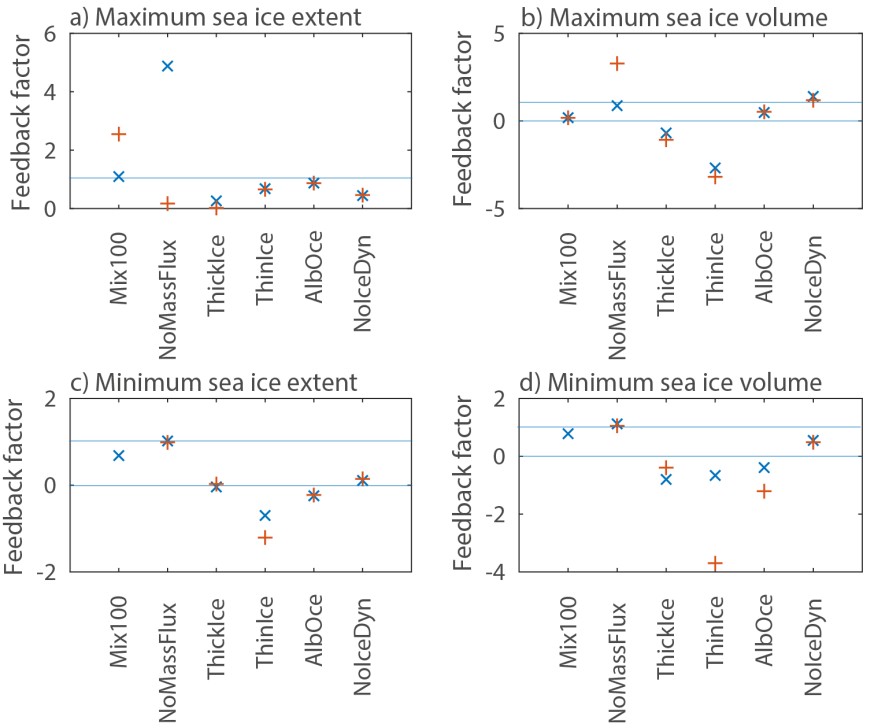


Figure 7. Atmospheric feedback factor for experiments starting in March (blue x) and September
(red +) for a) the maximum sea ice extent, b) maximum sea ice volume, c) minimum sea ice extent and
d) minimum sea ice volume. Light blue lines are drawn at values of 0 and 1 (with positive feedback
between those two lines). The equivalent figure for the feedback gain $G$ is given as Fig. S4.



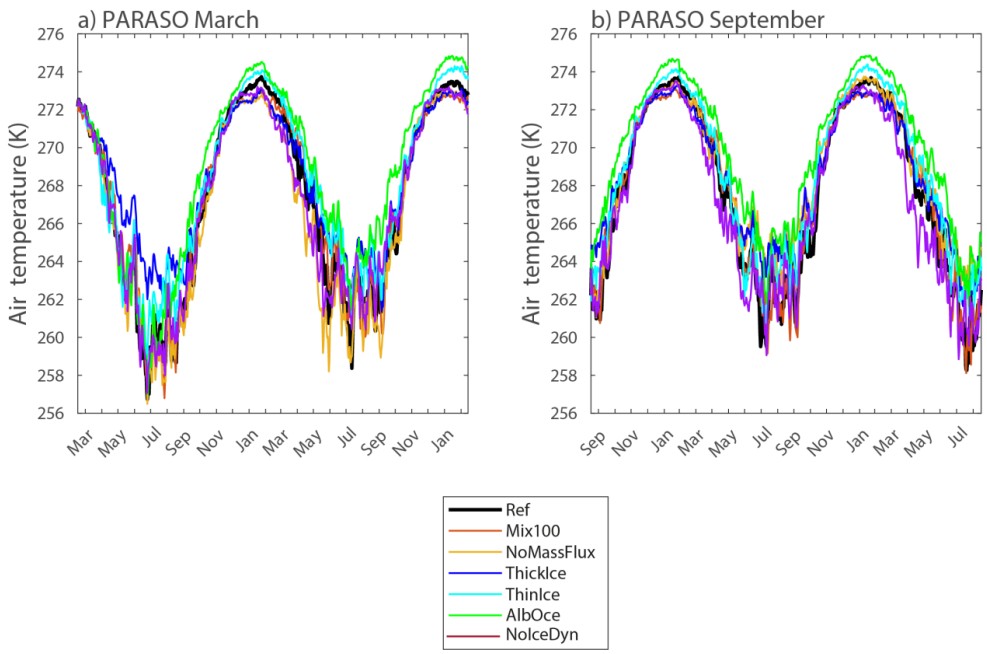

Figure 8. Surface air temperature (in K) averaged over the oceanic region south of 60°S in the group of experiments starting a) in March and b) in September for the PARASO configuration.

## 5  Discussion and conclusions

We have performed a series of 24 sensitivity experiments to analyze the role of key sea ice processes and coupling mechanisms between sea ice, ocean and atmosphere in driving the seasonal cycle of the Antarctic sea ice extent. In order to obtain clear signals and identify the mechanisms at play, deliberately strong and idealized perturbations have been used in our simulations. One limiting aspect arising from making such a design choice is the resulting lack of ability to directly compare the experiments with observational datasets. Furthermore, our quantitative results may be model-dependent, as they can be influenced by the way physical processes are represented in the models and by the biases in the model mean state, which can have a strong influence on the response of models to perturbations (e.g. Goosse et al., 2018; Massonnet et al., 2018). Additionally, the experimental design itself may have an impact on the way some of the processes are evaluated. However, we consider that the relative importance of the different processes and their description are robust and we will thus focus on those aspects here.

Recall that all the simulations used the same atmospheric forcing (for NEMO simulations) or the same conditions at the boundaries of the domain of the regional atmospheric model that significantly constrain the seasonal evolution of the sea ice (PARASO simulations). Changes in the large-scale atmospheric conditions or in the passage of synoptic storms close to the ice edge, for instance, are known to have a strong impact on the evolution of the ice extent (e.g., Handcock and Raphael, 2020). While this role of the atmospheric variability is not addressed here, the analyses of the processes at play could provide insight for understanding how the ice-ocean system responds to interannual variations of the atmospheric conditions. For instance, our results are consistent with the large role



attributed to the sea ice dynamics and thus to the interannual variability in winds in driving changes in
sea ice extent anomalies during the retreat season (e.g., Kusahara et al., 2019; Eayrs et al., 2020)
Our experiments are too idealized to provide explicit recommendations for model improvements
but the identification of the key processes can help to target the changes that might have the largest
impact. In particular, the delayed onset of the seasonal sea ice retreat after the maximum present in
our simulations can possibly be related to a too thick ice cover, which may be associated with a
misrepresentation of processes in the marginal ice zone (Roach et al., 2018, 2019; Alberello et al., 2020;
Horvat, 2021).
We have focused on the sea ice extent integrated over the whole Southern Ocean, although the
net influence of a process may be the result of opposite effects between sectors of the Southern Ocean
or between coastal regions and the open ocean. For instance, removing ice dynamics tends to increase
the ice thickness close to the coast and decrease it at the sea ice edge because of a reduced ice
transport, with a clear impact on the temperature changes. This is an illustration that our conclusions
derived for the whole ice pack are not necessarily valid for a specific region.
Overall, our results confirm the earlier finding that the model physics have only a moderate effect
on the timings of the maximum and minimum Antarctic sea ice extents, which are rather controlled by
the insolation cycle (Roach et al., 2022). Deactivating the sea ice dynamics in our models induces an
earlier maximum and a tendency towards a later minimum, but the shift is at maximum of the order
of one week or two, which is within the range of year-to-year fluctuations in the observed record.
Thicker ice can delay the maximum and a lower albedo lead to an earlier minimum, but similarly this
does not strongly modify the shape of the seasonal cycle, in particular its asymmetry. Our experiments
are only 2 years in length and there is a possibility that the shifts would become larger at equilibrium,
but in the experiments featuring a clear drift (such as NoIceDyn_PAR and AlbOce_PAR), we observe a
change in the values of the maximum and minimum ice extents from the first to the second year rather
than on their timing. The only exception is related to strong open ocean convection that can stop the
ice advance season efficiently when it is triggered in the model.
Nevertheless, our results demonstrate that sea ice physics and interactions with the atmosphere
and ocean control many other aspects of the seasonal cycle of the ice extent, such as the values of the
maximum and minimum and the speed of the retreat. They thus strongly modulate the overall impact
of the sea ice in the climate system, in particular on the radiative balance through the modification of
the surface albedo and on the exchanges of heat and carbon between the ocean and atmosphere.
Our sensitivity experiments have also illustrated clear distinctions between the dynamics of the
sea ice advance and retreat seasons. The sea ice extent advance from March to August is nearly
insensitive to the perturbations applied, with nearly identical evolution of the sea ice extent in our
experiment over this period if they start from the same initial conditions in March. If the conditions
are different in March (e.g., inherited from differences during the previous melting season), this has
an effect during the whole advance season. We can interpret those results in the following way. The
very weak incoming solar radiation between March and August imposes a large heat loss over the
Southern Ocean and the response of the system depends more on the heat available in March (and
thus of conditions at that time) than on any other element in the system. However, the sea ice
processes during the ice advance season can have an indirect effect by changing the sea ice thickness
and modifying the sea ice extent later in the year. This is the case for the ice production-entrainment
feedback that limits the ice growth in winter. During the ice advance season, this has no major impact
on the ice extent itself as it modulates the characteristics of sea ice that is already present, but the
modification of the thickness has an influence later during the retreat.
The timing of the beginning of the seasonal sea ice retreat and its rate also depend on the late
winter conditions, with thicker ice melting later. However, the retreat rate differs strongly between
the experiments, and this may have a larger impact on the spring and summer ice extents than the
conditions in September. Among all the processes influencing the retreat rate, the ice albedo feedback
is the dominant one, with a lower albedo, whether it is induced directly by a change in albedo (AlbOce)
or indirectly by a thinner ice (ThinIce) that melts faster, strongly accelerating the ice retreat. The ice
transport also plays a clear role by transporting sea ice northward where it melts. Neglecting this
process therefore leads to a large increase in summer ice extent. This larger dependence on several
key physical processes during the seasonal ice retreat is consistent with the larger climate model
sensitivity to changes in parameters in spring and early summer than during the ice advance season
(e.g., Urrego-Blancoet al., 2016; Schroeter and Sandery, 2022) and with the larger interannual
variability in the melt rates observed over the satellite period than in the growth rates (e.g., Eayrs et
al., 2020).
From a prediction point of view, the findings of this paper are also consistent with the idea that
the seasonal predictability of Antarctic sea ice extent depends on the season itself (Chevallier et al.,
2019; Marchi et al., 2020). A diagnostic predictability study using satellite data has revealed that
February is the month for which the sea ice extent anomalies exhibit the largest autocorrelations for
all lead times up to 55 days (Chevallier et al., 2019). This is in line with our findings showing that the
seasonal development of sea ice extent during the growing season is minimally controlled by physics
but rather by insolation and initial conditions. By contrast, the lowest autocorrelations of sea ice extent
anomalies are reached in the melting season, with complete loss of predictability in mid-November.
This is again in line with our results that multiple physical factors control the dynamics of sea ice melt.
The impact of all the sea ice and oceanic processes investigated here on the ice extent in winter
are amplified by the coupling with the atmosphere and our experimental design allow us to quantify
this amplification. The largest winter atmospheric feedback occurs for perturbations in albedo, as this
strongly modifies atmospheric temperature and humidity, amplifying the response of the ice. The
effect of the ice production-entrainment feedback is also strongly amplified by the atmospheric
coupling, as it brings thermal energy to the surface that melts ice but also warm up the atmosphere,
increasing the response of sea ice. By contrast, negative atmospheric feedbacks can develop for the
ice thickness and volume. In particular, larger heat losses due to higher conductive heat fluxes through
the sea ice can lead to greater sea ice formation. This induces a larger thermal energy transfer from
the ice-ocean system to the atmosphere that reduces the initial heat loss, resulting in a negative
atmospheric feedback on the thickness and potentially on the summer extent.
Roach et al. (2022) identified the role of insolation in controlling the observed asymmetry in the
growing and melting of Antarctic sea ice. Our idealized sensitivity experiments show that within this
robust cycle, the melt rate and maximum and minimum sea ice extents can be affected by sea ice-
ocean exchanges, sea ice processes, and ice dynamics. We also demonstrated quantitatively how
atmospheric feedback can enhance the effect of perturbations, but also in some cases damp it.
Although it is an idealized study, it highlights the major role of albedo and sea ice transport in the sea
ice extent seasonal cycle and as key processes to target in model development and process
understanding.
**Code and data availability**
As described in detail in Pelletier et al. (2022a), the PARASO sources can be obtained by CLM-
Community members on their RedC (https://redc.clm-community.eu/ then "COSMO-CLM" then
"Downloads"). All PARASO sources, except the COSMO routines, are publicly available for didactic
purposes at https://doi.org/10.5281/zenodo.5576201 (Pelletier et al., 2021) as well as the files to run
the model in the same configuration as here (Pelletier and Helsen, 2021). The NEMO3.6 version is
available from https://forge.ipsl.jussieu.fr/nemo/browser/branches/UKMO (Mathiot and Storkey,
730    2018).

**Supplement link**:
Supplementary information is available as a separate file.
**Author contributions.** HG initiated the study and designed the sensitivity experiments after
discussions with all the co-authors. FK performed the simulations. FK and PVH prepared the model
outputs for the analyses. HG made the analyses and the figures and all the co-authors contribute in
the interpretation of the results. HG wrote the manuscript, with inputs from all co-authors
**Competing interests**:
The authors declare that they have no conflict of interest.
**Acknowledgements.** This work was performed in the framework of the PARAMOUR project,
"Decadal predictability and variability of polar climate: the role of atmosphere-ocean-cryosphere
multiscale interactions", supported by the Fonds de la Recherche Scientifique – FNRS and the FWO
under the Excellence of Science (EOS) program (grant no. O0100718F, EOS ID no. 30454083). The
computational resources were provided by the VSC (Flemish Supercomputer Center), funded by the
Research Foundation Flanders (FWO) and the Flemish Government, the Center for High Performance
Computing and Mass Storage (CISM) of the Université catholique de Louvain (CISM/UCL) and the
Consortium des Équipements de Calcul Intensif en FédérationWallonie Bruxelles (CÉCI), funded by the
Fond de la Recherche Scientifique de Belgique (F.R.S.-FNRS) under convention 2.5020.11 and by the
Walloon region. HG is research director with the F.R.S.-FNRS. EBW was supported by the Office of
Naval Research-DRI grant N00014-18-1-2175. LR was supported by the National Oceanic and
Atmospheric Administration (NOAA) Climate and Global Change Postdoctoral Fellowship Program,
which is administered by UCAR's Cooperative Programs for the Advancement of Earth System Science
(CPAESS) under award NA18NWS4620043B.



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
