# Peer review of "Modulation of the seasonal cycle of the Antarctic sea ice extent by sea ice"

_The Cryosphere, 2022_

## Author Comment (AC1)

Answers to the comments of reviewer 1.

*In reviewing this manuscript, I looked at the rationale for the research, the method of study (I did not evaluate the models themselves) and the interpretation of the results.*

*This is article describes a careful attempt to isolate factors that influence the seasonal cycle of Antarctic sea ice extent and to explain how they do so. The manuscript is well written. The research problem is clearly stated – "what, other than the cycle of insolation influences/controls the asymmetry of the seasonal cycle of Antarctic sea ice extent?". The goals of the study are clear as are the arguments supporting the need for the research and the links to already existing work.*

*The authors use a series of sensitivity modeling studies to determine the roles of the oceanic and atmospheric processes in the seasonal cycle of Antarctic sea ice. More specifically, they examine the sea ice (extent, volume, timing of advance and retreat and growth/melt rates) responses to changes in the mixed layer depth (and the implied impact on heat storage), sea ice thickness, surface albedo, and ice dynamics. These simulations are short and there are caveats, but these are clearly stated, and results are interpreted within the bounds of these caveats. Even with these constraints, the results allow a better understanding of how the sea ice responds to different processes and the role of the atmosphere.*

*Overall, this study is immediately valuable to the field. It is, to my knowledge, the first of its kind to try to assess the response of the seasonal cycle to these key processes. Of course, sensitivity studies that involve a longer set of simulations may give more (statistically) reliable results, but these initial results seem physically sound and have great potential for interpreting and understanding the variability seen in observations sea ice extent around Antarctica.*

We would like to thank the reviewer for the positive evaluation and the helpful comments. Our responses are in blue, after the comments of the reviewer, which are in italics. The suggestions of modified text in the revised version are in green.

*I have only a few minor comments/suggestions to make. They follow.*

Line 205/206: How valid is this assumption - biases are small enough to have only a marginal effect on the response to the perturbation?

This is a very interesting point and a question hard to answer. The validity of the assumption may be evaluated by comparing the results of different models or different model versions and determining how the different biases in those models impact the results. This is unfortunately a lot of additional work and, even in this case, isolating the impact of the biases in the results is not easy. This would thus require a specific study. Nevertheless, the sentence included in the submitted version was a bit short. We know that the mean state influences the response to a perturbation (see for instance the discussion in Massonnet et al. 2018) and the evaluation of feedbacks (e.g., Goosse et al. 2018), as mentioned in the first paragraph of section 5. We also discuss in section 4 how the biases in the estimate of summer sea ice extent influences the quantification of atmospheric feedbacks during this season. We thus propose to modify the sentence to expand a bit the discussion on this point:

Each sensitivity experiment will be compared to the reference simulation using the same model configuration and initial state. This standard method implicitly assumes that the biases remain nearly constant in those pairs of experiments and the effect of those biases on the quantification of the response to the perturbation is largely removed by performing the difference between the experiments. However, even with this procedure, the biases can still have in some cases a clear impact on the quantification of feedbacks, as discussed in section 5 for the summer sea ice extent.

*Line 229/230 - Are you saying that this (assuming that sea ice salinity is the same as the ocean surface salinity) is what you did in the model? I assume yes. So, make this an active statement.*

On a practical point of view, we put to zero the mass fluxes at the sea-ice ocean interface in the sensitivity experiment. This is easier as sea ice and ocean salinities are variables in the model. To make it more explicit, we propose to replace the sentence by:

In practice, we thus set all the mass fluxes at the sea ice -ocean interface to zero in NoMassFlux but this is equivalent to assuming that sea ice salinity is the same as the ocean surface salinity.

Line 237/238 – I had to read several times to make sure that I understood what you meant. This, way of writing makes it a little confusing.  Can you redraft for clarity?

We propose to remove the parentheses and change the sentence to:

This is achieved by increasing the thermal conductivities of the ice and snow by a factor of five in ThickIce and by decreasing the thermal conductivities of the ice and snow by a factor of five in ThinIce.

Line 256/257 - This assumption might very well be valid but here the atmospheric feedbacks focus on the heat exchanges.  Can you make any comment on the effect that the lack of dynamics associated with atmospheric motion might have on your simulations?

The perturbation of surface conditions obtained in the sensitivity experiment can also have an impact on the dynamics and on the atmospheric circulation simulated by the regional atmospheric model COSMO-CLM. This is included in our evaluation of the atmospheric feedbacks that is based on an overall comparison of a configuration of NEMO that is forced and another configuration in which NEMO is coupled to the regional atmospheric model. However, as the dynamical response is expected to be more difficult to isolate in our short experiments that use boundary conditions that constrain the large-scale circulation, this potential dynamical response is not investigated in detail in the manuscript. We suggest to make this more explicit in the revised version by adding in the text that we will focus on the processes associated with surface heat exchanges but we not investigate in detail the potential changes in the winds and atmospheric dynamics in response to the perturbation:

While the perturbations can potentially influence the atmospheric dynamics, and thus winds for instance, we will focus on the feedbacks on heat exchanges at the surface as they are more directly impacted in the sensitivity experiments.

Lines 375 -383 – It is worth it to include a Figure reference here to aid the reader.

We will add a reference to Figures 4 and 2 in the revised version.

---

## Author Comment (AC2)

Answers to the comments of reviewer 2.

*General comments:*

*The seasonal cycle of Antarctic sea ice extent is greatly asymmetric, with a much slower phase of ice advance in austral fall than the rapid ice retreat in spring. Understanding this asymmetry in Antarctic sea ice extent is of fundamental importance, as it provides insight into processes linking the ice with the ocean and the atmosphere. As discussed in the manuscript, a substantial advancement has been made recently (Roach et al., 2022), showing that the asymmetry is ultimately induced by the incoming solar radiation. However, the modulation of the asymmetry by other processes that could be responsible, for example, for interannual variations in the asymmetry, remain poorly understood.*

*The study by Goosse et al. provides novel insight into this process by exploring the role of sea ice processes (dynamics and thermodynamics) and exchange processes with the ocean and atmosphere in driving a modulation of the seasonal asymmetry of the Antarctic sea ice cycle. They use a suite of model experiments to show that ice anomalies during the period of ice advance are largely controlled by the initial conditions in summer and the thermodynamic growth of the ice induced by heat loss at the surface. In contrast, anomalies during the phase of ice retreat vary substantially depending on the processes that are altered in the model. In particular, the role of changes in surface albedo and sea ice transport stand out as processes controlling anomalies during the ice retreat.*

*This is a very timely and insightful study that is of importance to the sea ice community, but also the wider climate and modeling community. It is well written, and the conclusions seem robust and supported by the analysis presented. I only have few minor questions and comments listed below and I recommend the publication of this manuscript after they are addressed. I congratulate the authors on this interesting work and making a valuable contribution to the field.*

We would like to thank the reviewer for the positive evaluation and the helpful comments. Our responses are in blue, after the comments of the reviewer, which are in italics. The suggestions of modified text in the revised version are in green.

*Specific comments:*

*1)*

*A) P6L193ff: "This simulation thus has no interannual variability […]" -> I have difficulties understanding why this should be the case. I understand that there is no interannual variability in the forcing since it is simply one specific year applied consecutively. However, why shouldn't there be any interannual variability developing in the ocean or sea ice? While the forcing certainly imposes some constrained, the ocean and ice are still free running models that can develop a certain level of variability.*

Stating that the NEMO configuration has no interannual variability in the submitted version was a bit short. There is no variability in the forcing but, in theory, the model can indeed develop its own variability for sea ice and ocean. This is the case when analyzing daily fields at some locations, in particular when mesoscale oceanic eddies are present, where some differences can be found due to the internal dynamics in the model. Nevertheless, for the surface variables integrated in space or time such as the Antarctic sea ice extent, the contribution of the internal model variability is extremely small and totally negligible in the present framework. For this specific comment, we propose to simply change in the revised version:

'This simulation thus has no interannual variability' to 'The forcing thus has no interannual variability'

We then suggest to come back to the discussion of internal variability in NEMO and PARASO configurations later in the paragraph as proposed in response to the next comment.

B) P6L206: "in contrast to NEMO alone" -> As for the comment above, I wouldn't understand why that should be the case. Probably the interannual variability in NEMO is dampened compared to PARASO by the imposed atmospheric forcing.

Yes, as mentioned above, when the model is at its equilibrium, the interannual variability in NEMO is controlled by the forcing and the contribution of internal processes and therefore internal model variability are very weak. PARASO displays some internal variability because of the interactive atmosphere, but the magnitude is much lower than in a global coupled model for instance, because of the fixed conditions at the boundary of the domain. We propose to modify the revised version to make this point clearer:

Additionally, in NEMO alone, the model internal variability is very low for the surface variables analyzed in the present study because of the strong constraint provided by the atmospheric forcing. Due to the inclusion of an interactive atmosphere, PARASO can develop some internal variability within its domain despite the fixed condition imposed at the boundaries.

C) P16-18: If I understand correctly the approach presented in this section assumes that there is no interannual variability in the ocean and ice in the NEMO simulation. Given my comment above, I wonder how this calculation and the conclusions of this section were affected if indeed there was some interannual variability in the ocean and ice in NEMO?

This is a very interesting point, but unfortunately it is hard to answer. That would be very instructive to make similar experiments for various years or conditions to quantify the role of the feedbacks in each case and see how the value of the feedback parameters change, as a function of the forcing itself or of the mean state of the system. We can imagine for instance that the feedbacks with ice dynamics are larger for years with stronger winds and larger ice transport. We provide here a first order of magnitude for the feedbacks investigated. Refining this estimate to determine how the values obtained change with conditions and for different years would be a very nice extension of our work, but we can only mention it as a perspective at this stage. We suggest to add this perspective at the end of the second paragraph of the conclusion where interannual variability is already discussed (but in a different context):

Conversely, the magnitude and relative importance of the different processes and feedbacks investigated in this study may vary from one year to another, as a function of the state of the system or of the large-scale forcing (e.g. Goosse et al. 2018). It would thus be instructive to repeat the analyses performed here for various years and conditions to determine how this affects the value of the feedback parameters.

*2) In some instances, I found it difficult to depict the signals described in the text from the figure because the lines appear close to each other. May I suggest to show the full seasonal cycles in Figure 1 as is, but for all other seasonal cycle plots (Figures 2, 3, 4, 8, and Supplementary) to show the anomalies with respect to the seasonal cycle. If the full seasonal is currently shown to distinguish the ice advance and retreat seasons that could be still retained by shading the background accordingly or similar. I think that would greatly help the reader to better identify the signals in the Figures.*

We decided to show in the submitted version the full seasonal cycle for all the experiments as it is generally not possible to see from the differences between the experiments if the perturbation

impacts the timing of the cycle (the timing of the maximum for instance or of the onset of retreat) or only its amplitude. Nevertheless, we agree that this choice may make the signals harder to see on the plots.

In the revised version, we propose to keep the full cycle for Figure 2 and Figure 4 as the timing of the ice extent evolution and ice growth/melt are important elements of our discussions but we would also provide a figure with the corresponding anomalies in the supplement (a shading did not really help to improve the clarity of the figure in the tests we made).

We consider that the curves for the ice volume are well separated from each other, so we propose to keep this figure as it is in the submitted version.

For Figure 8, as the timing of the evolution of temperature is similar in all experiments, we propose to show only the anomalies.

Following the same arguments, we propose to show only the full cycle for Figure S1 and S8 and only the anomalies for Figures S5 and S6.

3) P19L539-L540:

A) I guess here a reference to Figure S3 is missing.

The reference to Figure S3 will be added in the revised version.

*B) Looking at that figure, I was surprised that the overall ThinIce experiment shows a cooling in winter, whereas the cooling really is confined to a very narrow stretch along the coast and most of the rest of the pack ice region experiences a substantial warming. Given the large region of warming and the small region of cooling, I am wonder how it is possible to still get a cooling overall in Figure 8.*

We are sorry for the confusion. In the submitted version, we were mentioning a local cooling in ThinIce as seen on Figure S3, not an overall cooling that would be seen on Figure 8. The Referee is right that, integrated over the region south of 60°S, ThinIce is warmer than the reference experiment. This was hard to see on the submitted Figure 8 but this will be clearer on a figure displaying the anomalies in the revised version (see the answer to point 2 above). In addition to the new figure, we suggest to modify also the text to avoid confusion:

In ThinIce, the smaller heat conduction fluxes tend to induce an atmospheric cooling in winter but this effect is not strong enough to decrease the temperature in the majority of regions, likely because of a dominant effect of the albedo reduction in this experiment. However, a cooling is still found close to the continent (Fig. S3). As this is the region where the largest changes in sea ice thickness occur compared to the reference experiment, this dominates the effect of the coupling on the total ice volume. (For more information on the difference between the temperature responses in ThickIce and ThinIce, see the supplementary discussion).

C) Also, it seems counterintuitive to me why both ThickIce and ThinIce experience a warming for most of the pack ice region (Figure S3). Why would they respond in the same way, even if the perturbation is intended to have an opposing effect?

We were also intrigued by the behavior of the ThinIce experiment in comparison to ThickIce. We decided not to include a longer discussion on this in the submitted version as we considered that the manuscript was already heavy, with a lot of information. Following the question of the reviewer, we propose to add more details of this point in the supplementary material (referring to it in the main manuscript in section 4; see the proposed additional text for the point B above):

In ThickIce, a cooling is observed in summer and in regions in winter close to the ice edge compared to the reference experiment (Figure S3) due to the larger sea ice extent. However, this does not overwhelm the effect of the larger winter sea ice formation (Fig. 4) and thus the larger heat fluxes to the atmosphere within the pack that leads to an air temperature increase that dominates the regional mean (Fig. 8).

The opposite should occur in the ThinIce experiments. The lower sea ice formation (Fig. 4) and oceanic heat losses in ThinIce should lead to a cooling of the atmosphere within the ice pack, while the smaller ice extent should be associated with an atmospheric warming in the regions that are ice free in ThinIce and ice covered in the reference experiment. However, we find that the atmospheric warming due to a reduced ice extent expands to most of the pack in ThinIce, even in winter with cooling restricted to some regions close to the continent (Fig. S3). This extended warming is likely due to the strong changes in albedo and absorbed solar radiation in ThinIce (Fig. S4). The dominant role of the albedo is consistent with the generally colder temperatures in the first winter of ThinIce_PARA_Mar (Fig. 8), when the albedo effect did not yet have the time to act given that the experiments start at the end of summer. The larger fraction of leads within the ice pack also contributes to the warming in ThinIce.

4) P19L568-L604: In this section, I am wondering what role "noise" (on any time scale) in the simulations might play in leading to variations in gamma that are difficult to interpret as the ones described here. Couldn't the "unexplainable" variations in gamma also be due to the fact that there are no ensemble simulations run for these experiments?

Yes, we agree that the evaluation of gamma is impacted by the internal variability between the experiments. This potential impact of natural variability was mentioned at the end of the 'experimental design' in the submitted version but it is particularly important in the feedback framework as this framework implies to make the differences between several experiments and, when the signal is weak, those differences can be sensitive to uncertainties in the response to the perturbation due to the internal variability. We propose to recall specifically this limitation of our approach in section 4, when we explained that we will not discuss the feedback parameters when the changes are too small, mentioning that even when the signal is larger than the chosen threshold, some uncertainties due to the internal variability of the model remains in the evaluation of the feedback parameters.

We must recall here that we were not able to perform ensembles of simulations for our sensitivity experiments, leading to some uncertainties in the evaluation of the model response to the perturbations and thus in the estimate of the feedback parameters. Consequently, we have not analyzed the feedback factors when the coupled response is smaller than 0.2 million $km^2$ for sea ice extent or 0.2 thousand $km^3$ for sea ice volume, corresponding to very small changes in the system and large feedback factors (the coupled response appears in the denominator of $\gamma$). Consistently, we have focused the analyses on the second year of the experiments, as for the first year the changes in several experiments are too small. Nevertheless, even with those criteria, the low model internal variability still has an influence on the estimate of the feedback parameters and, in particular, it may also contribute to the non-linearities and values of $\gamma$ larger than one discussed below.

*5) P23L693-L644: I am wondering if the ice transport may also be pointed out here as a possible reason for large model biases during the retreat season, given that models do have substantial biases*

*in the transport and, as shown here, this process seems to be fundamental for anomalies during this season.*

Yes, we agree. We mention only one potential origin of the model biases in the submitted version but others are likely and the ice transport is certainly at the top of the list. We propose to add the following sentence in the revised manuscript:

Additionally, the too-fast retreat in our control runs is likely impacted by the model biases in the sea ice transport because of the dominant role this process during spring (e.g., Holland and Kwok 2012; Lecomte et al. 2016; Kusahara et al., 2019; Eayrs et al., 2020, Sun and Eisenman 2021).

*6) Section 5: Even though, as described, it is difficult to directly compare the simulations to the observed seasonal cycle in Antarctic sea ice and its changes due to the idealized nature of the experiments, I am wondering if anything could be learned from comparing the findings of this study with the findings by Holland (2014; https://doi.org/10.1002/2014GL060172). That study specifically investigated the seasonality of the observed sea ice trends, their relation, and some of the driving mechanisms.*

Thanks for pointing this study to our attention. The concepts of intensification and expansion (derivative of ice concentration and total ice area, respectively) introduced in this study provide very useful diagnostics. We do not introduce them formally here but we follow a similar approach for ice volume when analyzing sea ice growth and melt, which is the derivative of the ice volume (Figure 4), and when discussing changes from one season to next, which corresponds to expansion (but for sea ice extent) between the two seasons, in particular in the sub-section 'Minimum extent, subsequent advance season and amplitude of the seasonal cycle'. As mentioned, comparing the trends for the different seasons investigated in Holland (2014) and the processes controlling the mean seasonal cycle is not straightforward. Nevertheless, the role of winds in spring in explaining the trend and the impact of spring changes on the subsequent seasons is consistent with the processes investigated in our manuscript. We thus suggest to update in the revised version the discussion in the conclusion of the potential link between our results and the internal variability of the sea ice extent:

In particular, our results are consistent with the large role attributed to the sea ice dynamics, and thus to the interannual variability in winds, in driving changes in sea ice extent anomalies during the retreat season (e.g., Holland 2014; Kusahara et al., 2019; Eayrs et al., 2020) as well as with the impact of changes in spring on the sea ice extent trends observed in autumn (Holland 2014).

*7) Code and data availability: Is the model output available somewhere?*

The code is available as explained in the manuscript but the outputs correspond to more than 20 TB and it was not possible for us to make them available in an easy, self-descripting format on an open repository with a guarantee of long-term access. However, the outputs are available upon request without any restriction, except those technical ones.

*Technical corrections:*

*P2L30: "[…] configuration, including […]"*

This will be corrected in the revised version as suggested.

*P2L33: "This perturbation is […]"*

This will be modified in the revised version as suggested.

P3L58: Please use "—" instead of "-" (hyphen) for inserting the thought in this sentence and no spaces or commas before and after

This will be corrected in the revised version as suggested.

P5L154: "referred to as NEMO"

This will be corrected in the revised version as suggested.

P7L235-L253: This two paragraphs describe some underlying processes and it is not directly obvious to the reader if these are hypotheses or actual well understood processes in literature. In the latter case, some references would be helpful here.

The two paragraphs describe hypotheses about the role of specific processes that will be tested in the sensitivity experiments. Those hypotheses are based on our existing knowledge of the system. We propose to modify those two paragraphs in the revised version to make this clearer, adding citations to the relevant literature where appropriate.

P9L297-L300: Even after reading this sentence several times, I have troubles following the thought. Could you please split it up and simplify/clarify a little?

We propose to modify this sentence in the revised version as:

This impact of the ice thickness is well illustrated by the comparison between ThickIce_NEMO_Mar and Thin_NEMO_Mar, which have a difference of ice volume in winter of more than 20 $10^{12}$ m$^3$. ThickIce_NEMO_Mar has a maximum ice extent that is higher than in Thin_NEMO_Mar by 1.2 million km$^2$, a timing of the maximum extent delayed by 42 days compared to Thin_NEMO_Mar, and an extent that is larger than in Thin_NEMO_Mar by 3.3 million km$^2$ at the end of November (Fig. 2a).

P15L424, Figure 5 (and Supplementary): I assume that the purpose of showing these property vs. property scatter plots is to infer some sort of relation. If so, it would be good to also show the corresponding correlation coefficients and their statistical significance.

Yes, the idea of the scatter plot is to illustrate a relationship between properties in a more visual way than the tables (where a similar information is also available). However, we refrain ourselves to provide a correlation. We consider that each experiment is specific and, even though there are some common behaviors illustrated by the scatter plots, there are also some outliers as different processes are at play among the experiments. We also consider that estimating a statistical significance is not possible in the present framework. The different experiments were designed individually and could not be considered as being a reasonable sample of an ensemble or a population with similar characteristics. Furthermore, we do not see how we could estimate the effective size of our (already small) sample taking into account some possible dependency between the experiments. We prefer thus to keep the figures as illustrations rather than providing a number that may be misleading, keeping the quantitative evaluation for each experiment separately.

P16L441, Figure 6 (and Supplementary): I am afraid that the colorbar used in this figure is not an appropriate scientific colorbar and should not be used in scientific literature. For detailed reasons and tools to generate an appropriate colorbar in Matlab, please see for example this paper by Stauffer et al. (2015; https://doi.org/10.1175/BAMS-D-13-00155.1)

We will change the color bar for the revised version as suggested.

P18L530: "on average"

This will be corrected in the revised version as suggested.

P18L530: Lower case "s" for "south"

This will be corrected in the revised version as suggested.

*P19L536: I have difficulties seeing a more than 3K warmer atmosphere in the PARASO ThickIce experiment. Even when zooming in on the screen, it looks to me more like 1K, but it is difficult to say due to the scale at which this data is presented (see my comment 2 above).*

The peak difference in winter is larger than 5K the first year in ThickIce_PARA_MAR and higher than 3K in the second year of this experiment. Differences higher than 3K are also found in ThickIce_PARA_SEP during the two simulated winter. This will be clearer in the revised version on the plots of the difference between the sensitivity experiments and the control run (see the answer to the comment 2 above).

P19L553: "This interpretation is […]"

This will be modified in the revised version as suggested.

*P19L556/L560…and elsewhere: Not sure what "This" is referring to here. Please specify.*

L556. 'This' corresponds to the positive values mentioned in the previous phrases. It will be added in the revised version.

L560. 'This 'was referring to the larger response in NEMO. It will be modified in the revised version.

We will also check the revised version the text to clarify when 'This' is ambiguous and modify the text to remove the ambiguity.

P20L593-L592: Please remove "(" and ")"

This will be modified in the revised version as suggested.

P22L617: Please delete "key" as it is difficult to judge what it actually means

This will be modified in the revised version as suggested.

P24L705: "allows"

This will be corrected in the revised version as suggested.

P24L719: "dampen"?

This will be modified in the revised version as suggested.

References:

Stauffer, R., Mayr, G. J., Dabernig, M., & Zeileis, A. (2015). Somewhere Over the Rainbow: How to Make Effective Use of Colors in Meteorological Visualizations, Bulletin of the American Meteorological Society, 96(2), 203-216. https://doi.org/10.1175/BAMS-D-13-00155.1

Holland, P.R., Kwok R.: Wind-driven trends in Antarctic sea-ice drift. Nat Geosci 5(12):872–875. doi:10.1038/ngeo1627, 2012.

Holland, P. R., The seasonality of Antarctic sea ice trends, Geophys. Res. Lett., 41, 4230– 4237. doi:10.1002/2014GL060172, 2014.

Lecomte, O., Goosse, H., Fichefet, T., Holland, P. R., Uotila, P., Zunz, V.: Kimura, N., Impact of surface wind biases on the Antarctic sea ice concentration budget in climate models. Ocean Modelling 105, 60–70. https://doi.org/10.1016/j.ocemod.2016.08.001, 2016.

Sun S., Eisenman I.: Observed Antarctic sea ice expansion reproduced in a climate model after correcting biases in sea icedrift velocity. Nature Communications 12,1060, https://doi.org/10.1038/s41467-021-21412-z, 2021.